# GPS: Guided Positive Sampling in Self-Supervised Learning

## Abstract

This paper introduces *Guided Positive Sampling Self-Supervised Learning* (GPS-SSL), a method aimed at incorporating prior knowledge into Self-Supervised Learning (SSL) positive sample selection. Unlike current SSL methods relying solely on Data-Augmentations (DA) to generate positive samples, GPS-SSL creates a metric space aligning distances with semantic relationships and enabling informed positive sample selection through nearest neighbor sampling. A direct byproduct of GPS-SSL –and its core motivation– is the reduced importance of devising optimal DA recipes to learn performant representations. Since the proposed method solely alters the positive pair sampling, it can be coupled off-the-shelf with many SSL methods. Evaluation against baseline SSL methods on diverse datasets demonstrates the effectiveness of GPS-SSL, especially in scenarios with minimal DA; thus offering potential for further research on advancing SSL beyond careful DA design.

## 1 Introduction

Self-supervised learning (SSL) has recently shown to be one of the most effective learning paradigms across many data domains (Radford et al., 2021; Girdhar et al., 2023; Assran et al., 2023; Chen et al., 2020; Grill et al., 2020; Bardes et al., 2021; Balestriero et al., 2023). SSL belongs to the broad category of annotation-free representation learning approaches, which have enabled machine learning models to use abundant and easy-to-collect unlabeled data, facilitating the training of ever-growing deep neural network architectures.

Despite the SSL promise, current approaches require handcrafted a priori knowledge to learn useful representations. This a priori knowledge is often injected through the positive sample – *i.e.*, semantically related samples – generation strategies employed by SSL methods (Chen et al., 2020). In fact, SSL representations are learned so that such positive samples get as similar as possible in embedding space, all while preventing a collapse of the representation to simply predicting a constant for all inputs. The different strategies to achieve that goal lead to different flavors of SSL methods (Chen et al., 2020; Grill et al., 2020; Bardes et al., 2021; Zbontar et al., 2021; Chen & He, 2021). In computer vision, positive sample generation mostly involves sampling an image from the dataset, and applying multiple handcrafted and heavily tuned data augmentations (DAs) to it, such as rotations and random crops, which preserve the main content of the image.

The impact of designing DAs which are effective for the dataset at hand is enormous –as measured by its effect on performance (Garrido et al., 2023; Dangovski et al., 2021; Xiao et al., 2020; Tamkin et al., 2020; Kirichenko et al., 2023)–, to the point of producing a near random representation, in the worst case scenario (Balestriero et al., 2023). As such, tremendous time and resources have been devoted to designing optimal DA recipes, most notably for ubiquitous datasets such as ImageNet (Deng et al., 2009). From a practitioner's standpoint, positive sample generation could thus be considered solved if one were to deploy SSL methods *only* on such popular datasets. Unfortunately – and as we will thoroughly demonstrate throughout this paper –, common DA recipes used in those settings fail to transfer to other datasets. We hypothesize that as the dataset domains get semantically further from ImageNet, on which the current set of optimal DAs are designed, the effectiveness of DAs reduces. For example, since ImageNet consists of object-centric natural images focusing on $\sim 1000$ different object categories, we observe and report a reduction of performance on datasets consisting of more specialized images, such as hotel room images (Stylianou et al., 2019; Kamath et al., 2021), images of different types of airplanes (Maji et al., 2013), or medical images (Yang et al., 2023). Since searching for the optimal DAs is computationally intense (Tamkin et al., 2020), there

remains an important bottleneck when it comes to deploying SSL to new or under-studied domains. This becomes particularly noticeable when applying SSL methods on data gathered for real-world applications.

In this paper, we introduce a strategy to obtain positive samples, which generalizes the well established NNCLR SSL method (Dwibedi et al., 2021). While NNCLR proposes to obtain positive samples by leveraging known DAs and nearest neighbors in the embedding space of the network being trained, we propose to perform nearest neighbour search in the embedding space of a pre-defined mapping of each image to its possible positive samples. The mapping may generated by a clone of the network being trained – therefore recovering NNCLR – but perhaps most interestingly may also be generated by any available pre-trained network or be hand-crafted. This flexibility allows to (i) enable simple injection of prior knowledge into positive sampling –without relying on tuning the DA– and most importantly (ii) makes the underlying SSL method much more robust to under-tuned DAs parameters. By construction, the proposed method – coined GPS-SSL for Guided Positive Sampling Self-Supervised Learning–, can be coupled off-the-shelf with any SSL method used to learn representations, *e.g.*, BarlowTwins (Zbontar et al., 2021), SimCLR (Chen et al., 2020), BYOL (Grill et al., 2020), and VICReg (Bardes et al., 2021). We validate the proposed GPS-SSL approach on a benchmark suite of under-studied datasets, namely FGVCAircraft, PathMNIST, TissueMNIST, and show remarkable improvements over baseline SSL methods. We further evaluate our model on a real-world dataset, Revised-Hotel-ID (R-HID) (Feizi et al., 2022) and show clear improvements of our method compared the baseline SSL methods. Finally, we validate the approach on commonly used image datasets, i.e., Cifar10 and TinyImageNet, with known effective DAs recipes, and show that GPS remains competitive. Through comprehensive ablations, we show that GPS-SSL takes a step towards shifting the focus of designing *well-crafted DAs* to having a better *prior knowledge* embedding space in which choosing the nearest neighbour becomes an attractive positive sampling strategy.

The contributions of this paper can be summarized as:

- We propose a positive sampling strategy, GPS-SSL, that enables SSL models to use prior knowledge about the target-dataset to help with the learning process and reduce the reliance on carefully hand-crafted data augmentation recipes. The prior knowledge is a mapping between images and a few of their closest nearest neighbors that could be computed with a pre-trained network or even be hand-crafted.
- We evaluate GPS-SSL by coupling it with different SSL methods on a benchmark suite of understudied datasets. We show that GPS-augmented approaches significantly outperform the baseline methods when using minimal augmentations, highlighting the potential of GPS to learn representations from under-studied or real-world data. Moreover, when compared to SSL baselines leveraging strong augmentations or on well-studied datasets, the GPS-augmented approaches remain competitive.
- We further evaluate our model on datasets with under-studied applications of hotel retrieval which is of great importance to fight human trafficking. Similar to benchmark datasets, we see on this less studied dataset that GPS-SSL outperforms the baseline SSL methods by a significant margin.

We provide the code for GPS-SSL to reproduce our results on the (anonymized) GitHub: `https://anonymous.4open.science/r/gps-ssl-1E68`, for the research community.

## 2 Related Work

Self Supervised Learning (SSL) is a particular form of unsupervised learning methods in which a given Deep Neural Network (DNN) learns meaningful representations of their inputs without labels.

The variants of SSL are numerous. At the broader scale, SSL defines a pretext task on the input data and train themselves by solving the defined task. In SSL for computer vision, the pretext tasks generally involve creating different views of images and encoding both so that their embeddings are close to each other. However, that criteria alone would not be sufficient to learning meaningful representations as a degenerate solution is for the DNN to simply collapse all samples to a single embedding vector. As such, one needs to introduce an "anti-collapse" term. Different types of solutions have been proposed for this issue, splitting SSL methods into multiple groups, three of which are: 1) Contrastive(Chen et al., 2020; Dwibedi et al., 2021; Kalantidis et al., 2020): this group of SSL methods prevent collapsing by considering all other images in a mini-batch as negative samples for the positive image pair and generally use the InfoNCE (Oord et al., 2018) loss function to push

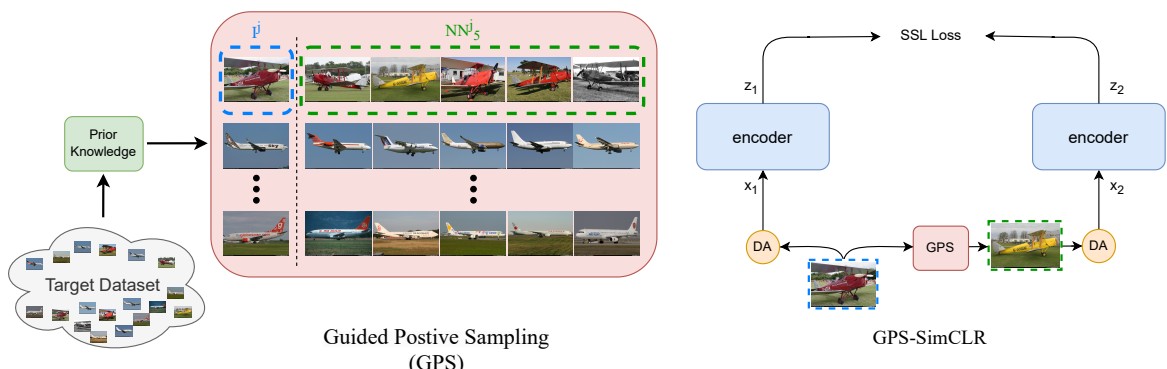

Figure 1: Our strategy, GPS-SSL, for positive sampling based on prior knowledge DA-based methods.

the negative embeddings away from the positive embeddings. 2) Distillation(Grill et al., 2020; He et al., 2020; Chen & He, 2021): these methods often have an asymmetric pair of encoders, one for each positive view, where one encoder (teacher) is the exponential moving average of the other encoder (student) and the loss only back-propagates through the student encoder. In general, this group prevents collapsing by creating asymmetry in the encoders and defines the pre-text task that the student encoder must predict the teach encoder's output embedding. 3) Feature Decorrelation(Bardes et al., 2021; Zbontar et al., 2021): These methods focus on the statistics of the embedding features generated by the encoders and defines a loss function to encourage the embeddings to have certain statistical features. By doing so, they explicitly force the generated embeddings not to collapse. For example, Bardes et al. (2021) encourages the features in the embeddings to have high variance, while being invariant to the augmentations and also having a low covariance among different features in the embeddings. Besides these groups, there are multiple other techniques for preventing collapsing, such as clustering methods (Caron et al., 2020; Xie et al., 2016), gradient analysis methods (Tao et al., 2022).

Although the techniques used for preventing collapse may differ among these groups of methods, they generally require the data augmentations to be chosen and tuned carefully in order to achieve high predictive performance (Chen et al., 2020). Although choosing the optimal data augmentations and hyper-parameters may be considered a solved problem for popular datasets such as Cifar10 (Krizhevsky et al., 2009) or ImageNet (Deng et al., 2009), the SSL dependency on DA remains their main limitation to be applied to large real-world datasets that are not akin natural images. Due to the importance of DA upon the DNN's representation quality, a few studies have attempted mitigation strategies. For example, Cabannes et al. (2023b) ties the impact of DA with the implicit prior of the DNN's architecture, suggesting that informed architecture may reduce the need for well designed DA although no practical answer was provided. Cabannes et al. (2023a) proposed to remove the need for DA at the cost of requiring an oracle to sample the positive samples from the original training set. Although not practical, this study brings a path to train SSL without DA. Also Van Gansbeke et al. (2020) proposes a two-stage learning process where first a clustering method with a pretext task is applied to the dataset and soft labels are acquired for performing an unsupervised learning on top of it. Additionally, a key limitation with DA lies in the need to be implemented and fast to produce. In fact, the strong DA strategies required by SSL are one of the main computational time bottleneck of current training pipelines (Bordes et al., 2023). Lastly, the over-reliance on DA may have serious fairness implications since, albeit in a supervised setting, DA was shown to impact the DNN's learned representation in favor of specific classes in the dataset (Balestriero et al., 2022; Kirichenko et al., 2023).

All in all, SSL would greatly benefit from a principled strategy to embed a priori knowledge into generating positive pairs that does not rely on DA. We propose a first step towards such Guided Positive Sampling (GPS) below.

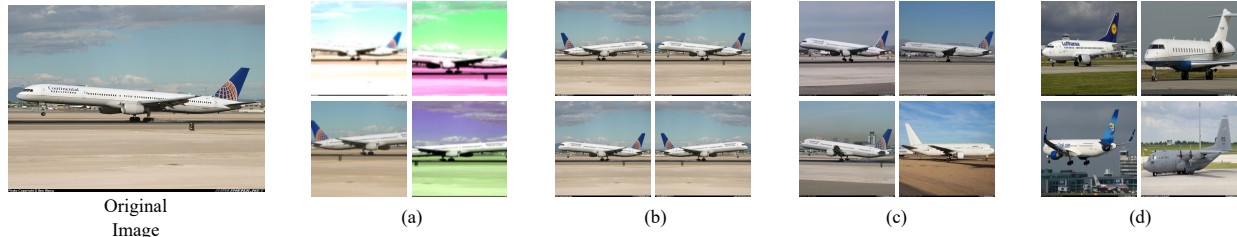

Original Image     (a)     (b)     (c)     (d)

Figure 2: An example (a) *StrongAug* and (b) *RHFlipAug* applied to an image from the FGVCAircraft dataset. Furthermore, (c) and (d) depict examples of the 4 nearest neighors calculated by CLIP and VAE embeddings, respectively.

# 3 Guided Positive Sampling for SSL

We propose a novel strategy, *Guided Positive Sampling Self-Supervised Learning* (GPS-SSL), that takes advantage of prior knowledge for positive sampling to make up for the sub-optimality of generating positive pairs solely from DA in SSL.

## 3.1 Nearest Neighbor Positive Sampling in Any Desired Embedded Space

As theoretically shown in several studies (HaoChen et al., 2021; Balestriero & LeCun, 2022; Kiani et al., 2022), the principal factors that impact the quality of the learned representation resides in how the positive pairs are defined. In fact, we recall that in all generality, SSL losses, i.e., $\mathcal{L}_{\text{SSL}}$, that are minimized can mostly be expressed as

$$\mathcal{L}_{\text{SSL}} = \sum_{(\boldsymbol{x}, \boldsymbol{x}') \in \text{PositivePairs}} \text{Distance}(f_\theta(\boldsymbol{x}), f_\theta(\boldsymbol{x}')) - \text{Diversity}(\{f_\theta(\boldsymbol{x}), \boldsymbol{x} \in \mathbb{X}\}), \tag{1}$$

for the SSL network $f_\theta$ and current training or mini-batch $\mathbb{X}$, with a distance measure such as the $\ell_2$ norm or the cosine similarity, and a diversity measure such that the rank of the embeddings or proxies of their entropy. All in all, defining the right set of PositivePairs is what determines the ability of the final representation to solve downstream tasks. The common solution is to repeatedly apply DA onto a single datum to generate such positive pairs:

$$\text{PositivePairs} \triangleq \{(\text{DA}(\boldsymbol{x}), \text{DA}(\boldsymbol{x})), \forall \boldsymbol{x} \in \mathbb{X}\}, \tag{2}$$

where the DA operator includes the random realisation of the DA such as the amount of rotation or zoom being applied onto its input image. However, this strategy often reaches its limits since such DAs need to be easily implemented for the specific data being used, and needs to be known a priori. When considering an image dataset, the challenge of designing DA for less common datasets, e.g., FGVCAircraft, led practitioners to instead train the model on a dataset such as ImageNet, where strong DAs have already been discovered, and then transfer the model to other datasets. This however has its limits when considering images from completely different domains –e.g. medical images.

We propose GPS-SSL, an alternative strategy to sample positive pairs, which can be used off-the-shelf with any baseline SSL method –e.g., SimCLR, VICReg. GPS-SSL defines positive pairs through nearest neighbour sampling in an a priori known embedding space denoted as $g_\gamma$.

First, we define the collection of samples that are less than $\tau > 0$ away from a query sample $\boldsymbol{x} \in \mathbb{X}$ in the chosen embedding space as

$$\mathcal{B}(\boldsymbol{x}) \triangleq \{\boldsymbol{x}' \in \mathbb{X} : \|g_\gamma(\boldsymbol{x}) - g_\gamma(\boldsymbol{x}')\|_2^2 < \tau\}. \tag{3}$$

Note that $\tau$ could either be a constant or a function of $x$. In this study we employ the latter, where $\tau$ is defined based on the k-nearest neighbor distance of $x$. From Eq. (3), GPS-SSL obtains positive pairs by randomly selecting a point in $\mathcal{B}(\boldsymbol{x})$ as in

$$\text{PositivePairs}_{\text{GPS}} \triangleq \{(\text{DA}(\boldsymbol{x}), \text{DA}(\boldsymbol{x}')), \forall (\boldsymbol{x}, \boldsymbol{x}') \in \mathbb{X}^2 : \boldsymbol{x}' \in \mathcal{B}(\boldsymbol{x})\}. \tag{4}$$

In short, we replace the set of positive pairs generated from applying a given DA to a same input, by applying a given DA onto two different inputs found so that one is the nearest neighbor of the other in some embedding space provided by $g_\gamma$. From this, we obtain a first direct result below making GPS-SSL recover a powerful existing method known as NNCLR (Dwibedi et al., 2021).

**Proposition 1.** *For any employed DA, GPS-SSL which replaces Eq. (2) by Eq. (4) in any SSL loss (Eq. (1)) recovers (i) input space nearest neighbor positive sampling when $g_\gamma$ is the identity and $\tau \gg 0$, (ii) standard SSL when $g_\gamma$ is the identity but $\tau \to 0$, and (iii) NNCLR when $g_\gamma = f_\theta$ and $\tau \to 0$.*

The above result provides a first strong argument demonstrating how GPS-SSL does not reduce the capacity of SSL; in fact, it introduces a novel axis of freedom–namely the design of $(g_\gamma, \tau)$–to extend current SSL beyond what is amenable solely by tweaking the baseline SSL network $f_\theta$, or the used DA. The core motivation of the presented method is that the ability to design $g_\gamma$ reduces the laborious task of designing effective DA recipes. In fact, if we consider the case where the original DA is part of the original dataset

$$\forall \boldsymbol{x} \in \mathbb{X}, \exists \rho : DA(\boldsymbol{x}; \rho) \in \mathbb{X}, \tag{5}$$

i.e., for any sample in the training set $\mathbb{X}$, at least on DA configuration $\rho$ exists that produces another training set sample, with $\rho$ specifying the applied transformation; GPS-SSL can recover standard SSL albeit without employing any DA.

**Theorem 1.** *Performing standard SSL (employing Eq. (2) into Eq. (1)) with a given DA and a training set for which Eq. (5) holds, is equivalent to performing GPS-SSL (employing Eq. (4) into Eq. (1)) without any DA and by setting $g_\gamma$ to be invariant to that DA, i.e. $g_\gamma(DA(\boldsymbol{x})) = g_\gamma(\boldsymbol{x})$.*

By construction from Eq. (5) and assuming that one has the ability to design such an invariant $g_\gamma$, it is clear that the nearest neighbour within the training set for any $\boldsymbol{x} \in \mathbb{X}$ will be the corresponding samples $DA(\boldsymbol{x})$ therefore proving Theorem 1. That result is quite impractical but nevertheless provides a great motivation to GPS-SSL. Note that $g_\gamma$ not only has the ability to mitigate the DA design, but can also be used jointly with DA, hence allowing one to embed as much a priori knowledge as possible through both $g_\gamma$ and said DA simultaneously.

**The design of $g_\gamma$.** The proposed strategy (Eq. (4)) is based on finding the nearest neighbors of different candidate inputs in a given embedding space. There are multiple ways for acquiring an informative embedding space, i.e., a prescribed mapping $g_\gamma$. Throughout our study, we will focus on the most direct solution of employing a previously pre-trained mapping. The pre-training may or may not have occurred on the same dataset being considered for SSL. Naturally, the alignment between both datasets affects the quality and reliability of the embeddings. If one does not have access to such pre-trained models, an alternative solution is to first learn abstracted representation of the data, e.g., using an MAE He et al. (2022) or VAE (Kingma & Welling, 2013), and then use the said representations for $g_\gamma$. In this setting, the motivation lies in the final SSL representation being superior to the encoder ($g_\gamma$) alone for solving downstream tasks.

We provide some examples of the resulting positive pairs with our strategy in Figure 1. In this figure, we use a pretrained model to calculate the set of $k$ nearest neighbors for each image $\boldsymbol{x}$ in the target dataset. Then for each image $\boldsymbol{x}$, the model randomly chooses the positive image from the nearest neighbors in embedding space (recall Eq. (4)). Finally, both the original image and the produced positive sample are augmented using the chosen DA and passed as a positive pair of images through the encoders. Note that as per Proposition 1, GPS-SSL may choose the image itself as its own positive sample, but the probability of it happening reduces as $\tau$ increases. As we will demonstrate in the later sections, the proposed positive sampling strategy often outperforms the baseline DA-based positive pair sampling strategy on multiple datasets.

**Relation to NNCLR.** The commonality of NNCLR and GPS-SSL has been brought forward in Proposition 1. In short, they both choose the nearest neighbor of input images as the positive sample. However, the embedding space in which the nearest neighbor is chosen is different. In NNCLR, the model being trained creates the embedding space which is thus updated at every training step, i.e., $g_\gamma = f_\theta$.

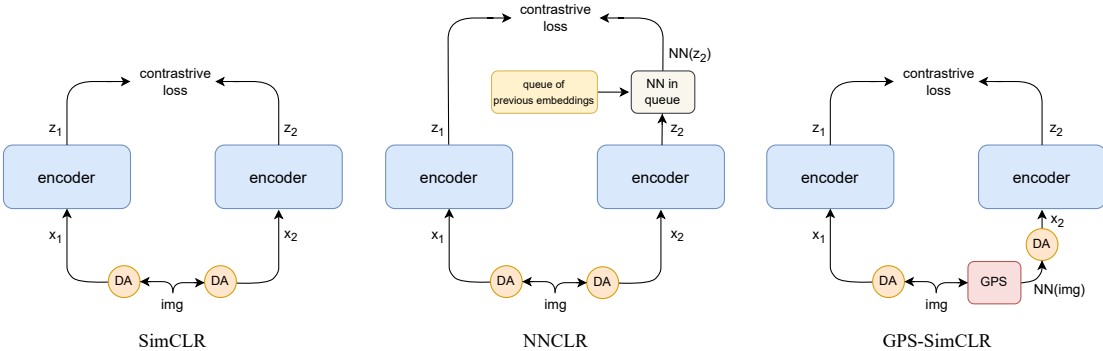

Figure 3: Architectures of SimCLR, NNCLR, and GPS-SimCLR. This figure demonstrates where the data augmentaiton (DA) happens in each method and also how the nearest neighbor (NN) search is different between NNCLR and GPS-SimCLR.

However, GPS-SSL generalizes that in the sense that the nearest neighbors can stem from any prescribed mapping, without the constraint that it is trained as part of the SSL training, or even that it takes the form of a DNN. The fact that NNCLR only considers the model being trained to obtain its positive samples also makes it heavily dependent on complex and strong augmentations to produce non degenerate results. Yet, our ability to prescribe other mappings for the nearest neighbor search makes GPS-SSL much less tied to the employed DA. We summarize and contrast with alternative SSL methods in Figure 3.

## 4 Empirical Validation on Benchmarked Datasets

In our experiments, we train the baseline SSL methods and the proposed GPS-SSL with two general sets of augmentations: *StrongAug*, which are augmentations that have been tuned on either Cifar10 (Krizhevsky et al., 2009) or on ImageNet in the case of TinyImageNet (Le & Yang, 2015), and the under-studied datasets (i.e., FGVCAircraft (Maji et al., 2013), PathMNIST (Yang et al., 2023), TissueMNIST (Yang et al., 2023), and R-HID). *RHFlipAug*, representing the scenario where we do not know the correct augmentations and use minimal ones. The set of *StrongAug* consists of random-resized cropping, random-horizontal flipping, color jittering, gray-scaling, Gaussian blurring, and solarizing, while *RHFlipAug* only uses random-horizontal flipping.

Table 1: Classification accuracy of a ResNet18 in different ablation settings.

(a) Comparing GPS-SimCLR when different pre-trained backbones (GPS-BB) are used to obtain embeddings for nearest-neighbor calculation, i.e., prior knowledge.

| **GPS-BB** | FGVCAircraft | |
|---|---|---|
| | *RHFlipAug* | *StrongAug* |
| ViT-B$_{\text{MAE}}$ | 10.53 | 29.55 |
| ViT-L$_{\text{MAE}}$ | 14.70 | 35.28 |
| RN50$_{\text{SUP}}$ | 18.15 | 41.47 |
| RN50$_{\text{VAE}}$ | 11.04 | 32.06 |
| RN50$_{\text{CLIP}}$ | 19.38 | 50.08 |
| ViT-B$_{\text{CLIP}}$ | **19.90** | **64.42** |

(b) Best performance in *StrongAug* setting of Sim-CLR and GPS-SimCLR given different learning rates (LR).

| **LR** | FGVCAircraft | |
|---|---|---|
| | SimCLR | GPS-SimCLR |
| 0.003 | 21.39 | 35.7 |
| 0.01 | 30.18 | 43.68 |
| 0.03 | 39.27 | 49.57 |
| 0.1 | 39.81 | **50.08** |
| 0.3 | **39.87** | 48.10 |

In order to thoroughly validate GPS-SSL as an all-purpose strategy for SSL, we consider SimCLR, BYOL, NNCLR, and VICReg as baseline SSL models, and for each of them, we consider the standard SSL positive pair generation (Eq. (2)) and the proposed one (Eq. (4)) by setting $\tau$ as a function of $x$ that is chosen

Table 2: Classification accuracy of baseline SSL methods with and without GPS-SSL on four datasets on *ResNet50* using pretrained RN50$_{CLIP}$ embeddings for positive sampling. We consider both *StrongAug* (Strong Augmentation) and *RHFlipAug* (Weak Augmentation) settings. The set of DA used for *StrongAug* are `random-resized-crop`, `random-horizontal-flip`, `color-jitter`, `gray-scale`, `gaussian-blur`, and `solarization`. For the *RHFlipAug* setting, the only DA used is `random horizontal flip`. We mark the **first**, second, and third best performing models accordingly. We can see that in the minimal augmentation setting, GPS-SSL improves the results significantly and boosts the performances to a comparable level to the strong augmentation setting for three of the datasets.

| Aug. | Method | Datasets | | | |
|---|---|---|---|---|---|
| | | Cifar10 (10 classes) | FGVCAircraft (100 classes) | PathMNIST (9 classes) | TissueMNIST (8 classes) |
| *RHFlipAug* | SimCLR | 47.01 | 5.61 | 63.42 | 50.35 |
| | BYOL | 41.79 | 6.63 | 67.08 | 48.00 |
| | NNCLR | 28.46 | 6.33 | 56.70 | 37.98 |
| | Barlow Twins | 41.73 | 5.34 | 53.27 | 43.57 |
| | VICReg | 37.51 | 6.18 | 46.46 | 39.79 |
| | GPS-SimCLR (ours) | 85.08 | 18.18 | 87.79 | 53.14 |
| | GPS-BYOL (ours) | 84.07 | 13.50 | 87.67 | 53.05 |
| | GPS-Barlow (ours) | 84.45 | 17.34 | 88.77 | **56.63** |
| | GPS-VICReg (ours) | **85.58** | **18.81** | **88.91** | 56.44 |
| *StrongAug* | SimCLR | 90.24 | 47.11 | **93.64** | 58.53 |
| | BYOL | 90.50 | 34.23 | 93.29 | 56.63 |
| | NNCLR | 90.03 | 34.80 | 92.87 | 52.57 |
| | Barlow Twins | 88.34 | 18.12 | 92.03 | **61.69** |
| | VICReg | **91.21** | 38.74 | 93.22 | 60.18 |
| | GPS-SimCLR (ours) | 91.17 | **55.60** | 92.30 | 55.59 |
| | GPS-BYOL (ours) | 91.15 | 44.28 | 92.40 | 55.03 |
| | GPS-Barlow (ours) | 88.52 | 20.23 | 91.98 | 57.04 |
| | GPS-VICReg (ours) | 89.71 | 47.29 | 92.55 | 55.79 |

from $x$'s k-nearest neighobr distances. We opted for a *randomly-initialized* ResNets backbone (He et al., 2016) as encoder. We also bring forward the fact that most SSL methods are generally trained on a large dataset for which strong DAs are known and well-tuned, such as ImageNet, and the learned representation is then transferred to solve tasks on smaller and domain-specific datasets. In many cases, training those SSL models directly on those less known datasets lead to catastrophic failures, as the optimal DAs have not yet been discovered. Lastly, we consider six different embeddings for $g_\gamma$: one obtained using supervised learning on ImageNet, two CLIP vision-language model (Radford et al., 2021) trained on LAION-400M Schuhmann et al. (2021), one with VAEs (Kingma & Welling, 2013) trained on Object365, and two with MAEs (He et al., 2022) trained on ImageNet as well. We also show that our method is more robust to hyper-parameter changes (see Tables 1a and 1b). Since the embedding models are trained on ImageNet or LAION-400M, the results reported throughout this study remain practical since the labels of the target datasets, on which SSL models are trained and evaluated, are never observed for the training of neither $g_\gamma$ nor $f_\theta$.

**Strong Augmentation Experiments.** The DAs in the *StrongAug* configuration consist of strong augmentations that usually distort the size, resolution, and color characteristics of the original image. First, we note that in this setting, GPS-SSL generally does not harm the performance of the baseline SSL model on common datasets, i.e. Cifar10 (Table 2). In fact, GPS-SSL performs comparable to the best-performing baseline SSL model on Cifar10, i.e., VICReg. We believe that the main reason lies in the fact that the employed DA has been specifically designed for those datasets (and ImageNet). However, we observe that GPS-SSL outperforms (on FGVCAircraft and Cifar10) or is comparable to (on PathMNIST) the baseline SSL methods for the under-studied and real-word datasets (Table 2). The reason for this is that, the optimal set and configuration of DA for one dataset is not necessarily the optimal set and configuration for another, and while SSL solely relies on DA for its positive samples, GPS-SSL is able to alleviate that dependency

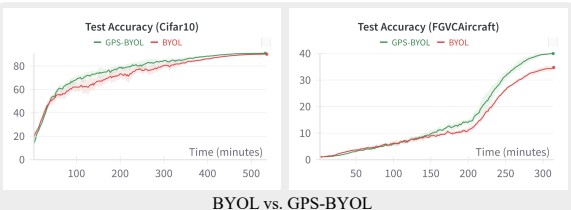

Figure 4: The impact of GPS-SSL on the runtime of SSL algorithms. We report the runtime of BYOL vs. GPS-BYOL and SimCLR vs. GPS-SimCLR on FGVCAircraft and Cifar10 for exactly 400 epoch. We see that the runtime of GPS-SSL remains very similar to the original baseline SSL method, whereas GPS-SSL improves the performance for the FGVCAircraft.

through $g_\gamma$ and uses positive samples that can be more useful than default DAs, as seen in Figure 2. Note that our method's runtime is similar to the baseline SSL method on the dataset it is learning and does not hinder the training process (Figure 4).

**Weak-Augmentation Experiments.** We perform all experiments under the *RHFlipAug* setting as well, showing GPS-SSL also produces high quality representations in this setting, validating Theorem 1. As shown in Table 2, GPS-SSL significantly outperforms all baseline SSL methods across both well-studied and under-studied datasets. These results show that our GPS-SSL strategy, though conceptually simple, coupled with the *RHFlipAug* setting, approximates strong augmentations used in the *StrongAug* configuration. This creates a significant advantage for GPS-SSL to be applied to real-world datasets where strong augmentations are not readily available, but where the invariances learned by $g_\gamma$ generalizes to them.

**Ablation Study** In this section we explore multiple ablation experiments in order to show GPS-SSL improves SSL and is indeed a future direction for improving SSL methods. First, we compare SSL and GPS training on Cifar10 and FGVCAircraft starting from a randomly initialized backbone (realistic setting), supervised ImageNet pre-trained weights, or CLIP pre-trained weights to explore whether the improvement of GPS-SSL are due to better positive sampling or simply because of using a strong prior knowledge. We show in Table 5 that GPS-SSL performs better than the baseline SSL methods, even when they both have access to the pre-trained network weights. This proves that the improvement in performance of GPS-SSL compared to baseline SSL methods is indeed due to better positive sampling.

Next, in Table 1a, we compare GPS-SimCLR with six different embeddings for $g_\gamma$. We observe that as the pre-trained embeddings become higher quality, the performance of our method increases in both the *RHFlipAug* and *StrongAug* setting. However, note that even given the weakest embeddings, i.e., the ViT-B$_{\text{MAE}}$ embeddings, GPS-SimCLR still outperforms the baseline SimCLR in the *RHFlipAug* setting, highlighting that the nearest neighbors add value to the learning process when DA recipes are not readily available.

We further explore if the improvement of GPS-SSL holds when methods are trained longer. To that end, we train a ResNet18 for 1000 epochs with SimCLR and VICReg with *StrongAug*, along with their GPS versions, on Cifar10 and FGVCAircraft and compare the results with the performance from 400 epochs. As seen in Table 3, the improvement of GPS-SSL compared to the baseline SSL method holds on FGVCAircraft dataset and remains comparable on Cifar10, showcasing the robustness of GPS-SSL.

Moreover, we compare GPS-SSL and baseline SSL methods on a larger scale dataset. We train and evaluate SimCLR and VICReg and their GPS versions with different $g_\gamma$ on TinyImageNet for 200 epochs with a ResNet50 under *StrongAug* and *RHFlipAug* settings. As seen in Table 4, GPS-SSL with the pre-trained backbone outperforms the SSL methods in both *RHFlipAug* and *StrongAug* settings. We can also see that even without a pre-trained backbone, GPS-SSL significantly outperforms SSL methods in *RHFlipAug* settings whereas the performance is comparable in the *StrongAug* setting.

Finally, we aim to measure the sensitivity of the performance of a baseline SSL method to a hyper-parameter, i.e., learning rate, with and without GPS-SSL. In this experiment, we report the best performance of

Table 3: Longer training: Test accuracy comparison of GPS-SSL after 1K epochs versus 400 training epochs. We observe the improvements of GPS-SimCLR are still significant when training for longer on FGVCAircraft and remain comparable on Cifar10.

| Method | Cifar10 | | FGVCAircraft | |
|---|---|---|---|---|
| | 400 eps | 1000 eps | 400 eps | 1000 eps |
| SimCLR | 88.26 | **91.25** | 39.87 | 45.55 |
| GPS-SimCLR | **89.57** | 91.10 | **50.08** | **51.64** |
| VICReg | 89.34 | **90.61** | 33.21 | 41.19 |
| GPS-VICReg | **89.68** | 89.84 | **45.48** | **49.29** |

Table 4: Scalability: Test accuracy comparison of GPS-SSL and baseline SSL on a *large scale* dataset: TinyImageNet (TinyIN). Results are reported after 200 training epochs with ResNet50. We also compare the setting when we train the backbone from scratch (ViT-L$_{\mathrm{MAE}}$ on TinyImageNet dataset) and when a pretrained model is used for the GPS backbone (RN50$_{\mathrm{CLIP}}$ pretrained on LAION-400M).

| Method | GPS | TinyIN | |
|---|---|---|---|
| | | *RHFlipAug* | *StrongAug* |
| SimCLR | — | 3.17 | 42.25 |
| VICReg | | 2.81 | 44.02 |
| GPS-SimCLR | scratch | 28.32 | 41.69 |
| GPS-VICReg | | 27.67 | 42.71 |
| GPS-SimCLR | pre-trained | **40.73** | 48.09 |
| GPS-VICReg | | 40.26 | **48.47** |

SimCLR and GPS-SimCLR given different learning rates in the *StrongAug* setting. We observe that GPS-SSL when applied to a baseline SSL method is as robust, if not more robust, to hyper-parameter changes. The results are reported in Table 1b. We perform further ablations, e.g., comparing GPS-SSL with the linear probing performance and trying additional GPS-BBs, in Appendix 7.5.

Table 5: Fine-tuning with GPS-SSL: Comparing SimCLR on ResNet50 with and without GPS-SimCLR from different model initializations with minimal (*RHFlipAug*) and strong (*StrongAug*) augmentations. RN50$_{\mathrm{RAND}}$, RN50$_{\mathrm{SUP}}$, and RN50$_{\mathrm{CLIP}}$ represents a ResNet50 with random weights (the standard setup), ImageNet supervised weights, and CLIP pretrained weights, respectively. We can see that GPS-SSL is also effective when fine-tuning the models.

| Method | Weight Init. | Cifar10 | | FGVCAircraft | |
|---|---|---|---|---|---|
| | | *RHFlipAug* | *StrongAug* | *RHFlipAug* | *StrongAug* |
| SimCLR | $RN50_{RAND}$ | 46.69 | 87.39 | 5.67 | 27.36 |
| GPS-SimCLR | | **85.2** | **90.48** | **17.91** | **43.56** |
| SimCLR | $RN50_{SUP}$ | 43.99 | 94.02 | 17.91 | 59.92 |
| GPS-SimCLR | | **91.3** | **95.53** | **39.45** | **66.88** |
| SimCLR | $RN50_{CLIP}$ | 45.57 | 90.26 | 6.21 | 41.04 |
| GPS-SimCLR | | **89.44** | **91.23** | **24.15** | **49.63** |

## 5   Case Study on the Hotels Image Dataset

In this section, we study how GPS-SSL compares to baseline SSL methods on an under-studied real-world dataset. We opt the R-HID (Feizi et al., 2022) dataset for our evaluation which gathers hotel images for the purpose of countering human-trafficking. R-HID provides a single train set alongside 4 evaluation sets, each with a different level of difficulty, ranging from known (seen) hotel chains and branches to unknown chains and branches.

Table 6: R@1 on different splits on R-HID Dataset for SSL methods. The splits are $\mathcal{D}_{SS}$: {branch: seen, chain: seen}, $\mathcal{D}_{SU}$: {branch: unseen, chain: seen}, $\mathcal{D}_{UU}$: {branch: unseen, chain: unseen}, and $\mathcal{D}_{??}$: {branch: unknown, chain: unknown}. We mark the best-performing score in **bold**.

| Method | $\mathcal{D}_{SS}$ | $\mathcal{D}_{SU}$ | $\mathcal{D}_{UU}$ | $\mathcal{D}_{??}$ |
|---|---|---|---|---|
| SimCLR | 3.28 | 16.76 | 20.30 | 16.00 |
| BYOL | 3.69 | 19.27 | 23.02 | 18.47 |
| Barlow Twins | 3.04 | 15.54 | 18.96 | 15.06 |
| VICReg | 3.41 | 17.52 | 20.45 | 16.53 |
| GPS-SimCLR | 4.84 | 23.67 | 26.30 | 22.28 |
| GPS-BYOL | 3.89 | 19.64 | 23.18 | 19.38 |
| GPS-Barlow | 4.49 | 21.98 | 25.23 | 20.82 |
| GPS-VICReg | **5.33** | **25.71** | **28.29** | **23.78** |

We evaluate the baseline SSL models with and without GPS-SSL with the R-HID dataset and report the Recall@1 (R@1) for the different splits introduced. Based on the findings from 2, we adapt the *StrongAug* setting along with the prior knowledge generated by a CLIP-pretrained ResNet50.

As seen in Table 6, SSL baselines always get an improvement when used with GPS-SSL. The reason the baseline SSL methods underperform compared to their GPS-SSL version is that the positive samples generated only using DA lack enough diversity since the images from R-HID dataset have various features and DA recipes limit the information the network learns; however, paired with GPS-SSL, we see a clear boost in performance across all different splits due to the information extracted from the nearest neighbors.

## 6   Conclusions

In this paper, we proposed GPS-SSL, a strategy to obtain positive samples for Self-Supervised Learning. In particular, GPS-SSL complements traditional DA-based positive sampling by producing positive samples from the nearest neighbors of the data as measured in a given embedding space. This approach provides an alternative axis of research in SSL methods that can be used alongside the design of DA recipes and losses, while remaining sensitive to the choice of embedding space. Through this strategy, we were able to train SSL methods on relatively under-explored datasets such as medical images–without having to search and tune for the right DA. Those results open new avenues to employ SSL on datasets for which effective DA recipes are not available. In fact, we observe that while GPS-SSL meets or surpasses SSL performances across our experiments, the performance gap is more significant when the optimal DAs are not known, e.g., in PathMNIST and TissueMNIST. Besides practical applications, GPS-SSL finally provides a novel strategy to embed prior knowledge into SSL and the more reliable the prior knowledge is with respect to the target dataset, the stronger GPS-SSL performs without merely depending on DAs.

**Limitations.** The main limitation of our method is akin to the one of SSL, it requires the knowledge of the embedding space in which positive samples are produced using the nearest neighbors. This limitation is on par with standard SSL's reliance on DA, but its formulation is somewhat dual (recall Theorem 1) in that one may know how to design such an embedding without knowing the appropriate DA for the dataset, and vice-versa. Furthermore, if the knowledge acquired is not compatible with the target dataset, GPS-SSL tends to underperform compared to SSL; this was seen in Table 2 for PathMNIST and TissueMNIST in the *StrongAug* setting. Alternative techniques like training separate and simple deep networks to provide such embeddings prior to the SSL learning could be considered for future research. More on the future work in Appendix 7.3.

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

# 7 Appendix

## 7.1 R-HID Splitting method

R-HID (Feizi et al., 2022) is created carefully to make sure no data leakage occurs. They mention how the total data is divided into the train and the multiple test splits. More specifically, first a set of chains (along with *all* their branches) are reserved for the $\mathcal{D}_{UU}$ to make sure the chains (super-classes) and branches (classes) are not seen during training. Next, out of the remaining chains, a set of the branches are chosen to add *all* of their images to the $\mathcal{D}_{SU}$ test split (since the training set will have other images from other branches from the same chain, but not the same branch images). Finally, out of the remaining branches, the images in each are split between $\mathcal{D}_{SS}$ and train, creating the final test split that has a subset of the branches seen during training. With this procedure, they make sure of the table of overlapping below. More details regarding the splits is provided in the original paper.

## 7.2 Computational and Memory Cost of GPS-SSL

The computational cost of GPS-SSL and standard SSL are not significantly different; the main distinction lies in the positive sampling and acquiring the positive samples. In our approach, we investigate acquiring the positive samples from a pre-trained network, which entails an additional cost of computing the embeddings of the target dataset and computing the $k$-nearest neighbors of each data point. However, it's important to note that the computation of the $k$-nearest neighbors needs to be performed only once during the training procedure and can be cached for any SSL method on the same dataset and GPS-BB, rendering the additional computation negligible.

Regarding additional memory usage, GPS-SSL requires storing the indexes of the $k$-nearest neighbors of each data point in the training set. Thus, the order of magnitude of the memory usage is $O(N)$ ($k \lll N$) for a dataset of size $N$. However, it's noteworthy that this array does not need to be loaded onto the GPU during training.

## 7.3 Hyper-Parameter Search

In all main experiments (Table 2), we train for both 400 nd 1000 epochs with a batch size of 256 using one RTX 8000 GPU for all methods. To ensure we are choosing the correct hyper-parameters for a fair comparison, we search over a vast range of hyper-parameter combinations ($lr \in \{1e^{-3}, 3e^{-3}, 3e^{-2}, 1e^{-2}, 3e^{-1}, 1e^{-1}, 1\}$, $classifier\_lr \in \{3e^{-2}, 1e^{-2}, 3e^{-1}, 1e^{-1}, 1, 3\}$, $weight\_decay \in \{1e^{-4}, 1e^{-3}\}$) and for GPS-SSL with all SSL baselines we also search over $k \in \{1, 4, 9, 49\}$). For experiments using *RHFlipAug* and *StrongAug*, we use nearest neighbors calculated based on embeddings created from a ResNet50 that have been CLIP pre-trained as the prior knowledge. Finally, for each method, we report the best classification accuracy for Cifar10, FGVCAircraft, PathMNIST, and TissueMNIST, and Recall@1 (R@1) for R-HID in Tables 2, 6, and 11. To calculate both metrics, we first train the encoder on the target dataset using the SSL method, with or without GPS-SSL. Then, for classification accuracy, we train a linear classifier on top of it, and for R@1, we encode all the images from the test set and calculate the percentage of images which their first nearest neighbor is from the same class.

## 7.4 Future Work

Given the broad applicability and efficiency of GPS-SSL in enhancing performance by integrating prior knowledge with data augmentations, we plan to extend this framework to other single-modal data domains as well as multi-modal domains. We anticipate that employing GPS-SSL on data modalities lacking well-defined data augmentations, such as language, could provide significant benefits. Additionally, incorporating such prior knowledge in training multi-modal networks represents a promising direction for future research. Furthermore, since all observations in this study were empirical, it would be valuable to pursue formal proofs for the propositions and theorems discussed and moreover the drawbacks of our method for different target datasets, thereby strengthening the theoretical foundation of our approach.

Table 7: Classification accuracy of baseline SSL methods with and without GPS-SSL on four datasets on *ResNet18* using pretrained $RN50_{CLIP}$ embeddings for positive sampling. We consider both *StrongAug* (Strong Augmentation) and *RHFlipAug* (Weak Augmentation) settings. The set of DA used for *StrongAug* are `random-resized-crop`, `random-horizontal-flip`, `color-jitter`, `gray-scale`, `gaussian-blur`, and `solarization`. For the *RHFlipAug* setting, the only DA used is `random horizontal flip`. We mark the **first**, second, and third best performing models accordingly.

| Aug. | Method | Datasets | | | |
|---|---|---|---|---|---|
| | | Cifar10 (10 classes) | FGVCAircraft (100 classes) | PathMNIST (9 classes) | TissueMNIST (8 classes) |
| *RHFlipAug* | SimCLR | 47.62 | 7.70 | 62.99 | 52.30 |
| | BYOL | 49.72 | 8.99 | 77.77 | 51.00 |
| | NNCLR | 71.74 | 8.10 | 56.92 | 42.59 |
| | Barlow Twins | 42.00 | 7.53 | 64.82 | 49.43 |
| | VICReg | 36.04 | 4.95 | 56.92 | 50.26 |
| | GPS-SimCLR (ours) | **85.83** | 18.48 | **88.62** | 55.98 |
| | GPS-BYOL (ours) | 84.56 | 14.79 | 81.66 | **56.21** |
| | GPS-Barlow (ours) | 84.83 | 18.12 | 87.79 | 55.86 |
| | GPS-VICReg (ours) | 85.38 | **20.16** | 87.83 | 55.26 |
| *StrongAug* | SimCLR | 88.26 | 39.87 | 91.56 | 61.51 |
| | BYOL | 86.90 | 27.33 | 91.24 | 60.73 |
| | NNCLR | 87.95 | 39.12 | 91.14 | 52.42 |
| | Barlow Twins | 88.89 | 25.71 | 92.23 | 60.06 |
| | VICReg | 89.34 | 33.21 | **92.27** | 59.41 |
| | GPS-SimCLR (ours) | 89.57 | **50.08** | 92.19 | 62.76 |
| | GPS-BYOL (ours) | 88.46 | 32.07 | 91.05 | 54.05 |
| | GPS-Barlow (ours) | 88.39 | 25.35 | 91.55 | **62.93** |
| | GPS-VICReg (ours) | **89.68** | 45.48 | 91.88 | 62.46 |

## 7.5 Ablation Study

### 7.5.1 Different Backbone

First, we provide the same experiments as in Table 2, but trained with a ResNet18 instead of a ResNet50 and provide the results in Table 7. We see the same results for ResNet50 (discussed for Table 2) also hold when ran on a smaller architecture, i.e., ResNet18. This shows the improvements of GPS-SSL over baseline SSL methods is more reliable and robust.

### 7.5.2 Additional Datasets

To further evaluate our method, we train and evaluate GPS-SSL on a large-scale dataset, i.e., ImageNet100, a fine-grained image classification dataset, i.e., Food101, and finally an additional image classification dataset, i.e., Cifar100. For ImageNet100, we train a ResNet50 for 1000 epochs and for the other two additional datasets, we train a ResNet18 for 400 epochs. As seen in Tables 8 and 9 GPS-SSL helps improve the performance of the original SSL methods in both *RHFlipAug* and *StrongAug* settings.

### 7.5.3 Ablating Different $k$s

We additionally ablate different values for $k$, i.e., the number of nearest neighbors for each datapoint to consider for the positive sampling selection while training a ResNet18 and ResNet50 on FGVCAircraft. Note that when $k = 0$, GPS-SimCLR reduces to the original SimCLR method. As seen in Figure 5, there

Table 8: Results of ResNet50 with SimCLR and GPS-SimCLR on a largescale dataset, e.g., ImageNet100, after training for more epochs.

| Method | GPS-BB | ImageNet100 | | | |
| | | 100 epochs | | 1000 epochs | |
| | | *RHFlipAug* | *StrongAug* | *RHFlipAug* | *StrongAug* |
|---|---|---|---|---|---|
| SimCLR | —— | 17.6 | 77.18 | 5.42 | 84.78 |
| GPS-SimCLR | RN50$_{CLIP}$ | **77.54** | **82.68** | **77.38** | **85.78** |
| GPS-SimCLR | ViT-L$_{MAE}$ | 70.02 | 77.84 | 71.50 | 83.18 |

Table 9: Results of ResNet18 with SimCLR and GPS-SimCLR on more largesclae datasets (Food101 and Cifar100).

| Method | GPS-BB | Food101 | | Cifar100 | |
| | | *RHFlipAug* | *StrongAug* | *RHFlipAug* | *StrongAug* |
|---|---|---|---|---|---|
| SimCLR | —— | 16.88 | 73.35 | 26.1 | 64.78 |
| GPS-SimCLR | RN50$_{CLIP}$ | **70.25** | **78.94** | **61.34** | **66.45** |

Table 10: Results of ResNet18 with MoCo v3 and GPS-MoCo v3 on Cifar10 and FGVCAircraft.

| Method | GPS-BB | Cifar10 | | FGVCAircraft | |
| | | *RHFlipAug* | *StrongAug* | *RHFlipAug* | *StrongAug* |
|---|---|---|---|---|---|
| MoCo v3 | —— | 49.7 | 87.9 | 8.37 | 32.7 |
| GPS-MoCo v3 | RN50$_{CLIP}$ | **83.27** | **90.39** | **15.60** | **45.87** |

is an optimal number of nearest neighbors to use and using more results in lower performance, yet still outperforming the original SimCLR.

### 7.5.4 MoCo v3

We also explore MoCo v3 (He et al., 2020; Chen et al., 2021), another SSL backbone based on momentum encoders, on FGVCAircraft and to do so, we train a ResNet18 for 400 epochs with and without GPS on MoCo v3. As seen in Table 10, GPS improves the performance of this method as well.

### 7.5.5 Finetuning for R-HID

We further try a trivial way of transferring knowledge from a pretrained network to other SSL baseline models and compare it to GPS-SimCLR; we initialize the base encoder in any SSL method, i.e., the ResNet18, to the pretrained network's weights, as opposed to random initialization, and train it i.e., finetuning. Ultimately, we compare the results on R-HID in Table 11.

Although this might perform better if the pretrained network was trained on a visually similar dataset to the target dataset, Table 11 shows that it may harm the generalization on datasets that are different, e.g., ImageNet and R-HID, compared to being trained from scratch. However, GPS-SSL proves to be a stable method for transferring knowledge even if the pretrained and target dataset are visually different (Table 6).

### 7.5.6 Comparing to Linear Probing

Finally, we compare the linear probing performance of the embeddings generated from different architectures, i.e. GPS backbones (GPS-BB), pretrained on different datasets, i.e., GPS Datasets (GPS-DS), with the performance of GPS-SSL using them. More specifically, in Tables 13 and 12, we compare the linear probe

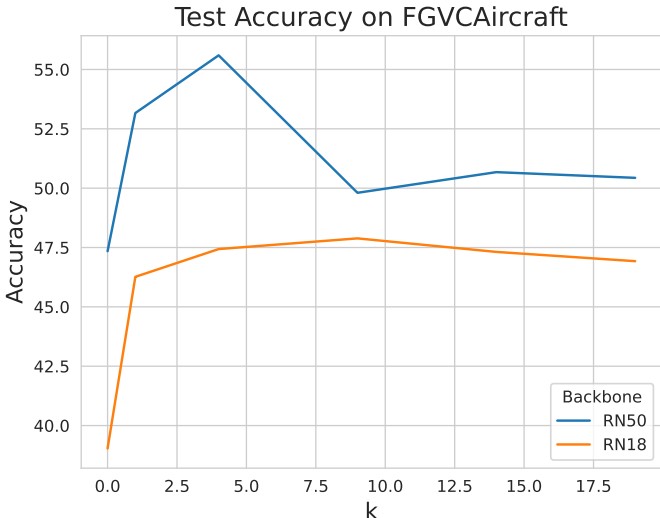

Figure 5: Ablation over different values of $k$ with GPS-SimCLR on FGVCAircraft.

Table 11: Comparing the R@1 performance of SSL methods on R-HID when trained from scratch against being finetuned, i.e., being initialized to a supervised ImageNet pretrained network, on a ResNet50. We highlight the difference in R@1 of the pretrained against the scratch version with green when it improves and red when it worsens.

| Method | Weight Init. | $\mathcal{D}_{SS}$ | $\mathcal{D}_{SU}$ | $\mathcal{D}_{UU}$ | $\mathcal{D}_{??}$ |
|---|---|---|---|---|---|
| **SimCLR** | $RN50_{RAND}$ | 3.23 | 16.10 | 19.62 | 15.12 |
| | $RN50_{SUP}$ | -0.10 | -0.21 | -0.40 | +0.27 |
| **BYOL** | $RN50_{RAND}$ | 3.27 | 16.25 | 20.20 | 15.91 |
| | $RN50_{SUP}$ | -0.57 | -1.75 | -2.23 | -1.50 |
| **NNCLR** | $RN50_{RAND}$ | 2.84 | 13.91 | 17.15 | 13.96 |
| | $RN50_{SUP}$ | -0.54 | -2.44 | -3.18 | -2.67 |
| **VICReg** | $RN50_{RAND}$ | 3.24 | 16.67 | 19.97 | 15.86 |
| | $RN50_{SUP}$ | -0.43 | -1.54 | -2.45 | -1.92 |

performance of the CLIP pretrained ResNet50 on LAION-400M (Schuhmann et al., 2021) along with vision transformers (ViTs) pretrained on ImageNet using Masked Auto Encoders (MAE) (He et al., 2022), a popular self-supervised method that also does not rely on strong augmentations. We see our method outperforms the linear probe accuracy of CLIP embeddings for both Cifar10 and FGVCAircraft and matches that of ViT-Base and ViT-Large for Cifar10 and ViT-Large for FGVCAircraft.

However, we further see that if we train the ViT-Large on the FGVCAircraft, using MAE with minimal augmentations, we can use that as the positive sampler for GPS-SSL and beat the baseline SSL method on FGVCAircraft. This shows that GPS-SSL does not entirely rely on huge pretrained models and that there is potential possibilities for training a positive sampler prior to applying GPS-SSL to further boost the performance of baseline SSL methods.

Table 12: Comparison of linear probing (LP) and GPS-VICReg's (with ResNet50) classification accuracy on FGVCAircraft with different GPS backbones (GPS-BB) pretrained with CLIP and masked auto encoders (MAE) on different datasets without supervision (GPS-DS). The performance of the vanilla VICReg is also depicted for comparison. RN50 and ViT-L refer to ResNet50 and ViT-Large, respectively.

| GPS-BB | GPS-DS | LP | GPS-VICReg | VICReg |
|---|---|---|---|---|
| $RN50_{CLIP}$ | LAION-400M | 44.55 | 46.44 | |
| $ViT-L_{MAE}$ | ImageNet | 37.32 | 38.44 | 39.99 |
| $ViT-L_{MAE}$ | FGVCAircraft | 17.01 | 42.87 | |

Table 13: Classification accuracy comparison of linear probing (LP) using embeddings with different GPS backbones (GPS-BB) pretrained with CLIP and masked autoencoders (MAE) on different upstream datasets, i.e., GPS-DS, and a trained ResNet50 with GPS-SimCLR on FGVCAircraft and Cifar10 using the same GPS backbones and datasets. RN50, ViT-L, and Vit-B refer to ResNet50, ViT-Large, and ViT-Base, respectively.

| GPS-BB | GPS-DS | Cifar10 | | FGVCAircraft | |
|---|---|---|---|---|---|
| | | LP | GPS-SimCLR | LP | GPS-SimCLR |
| $RN50_{CLIP}$ | LAION-400M | 87.85 | 91.17 | 44.55 | 53.81 |
| $ViT-B_{MAE}$ | ImageNet | 85.78 | 87.35 | 27.96 | 29.55 |
| $ViT-L_{MAE}$ | ImageNet | 91.45 | 90.11 | 37.29 | 35.28 |
| $ViT-L_{MAE}$ | FGVCAircraft | —— | —— | 17.01 | 46.93 |

### 7.5.7 Additional GPS-BB

As an additional and stronger GPS-BB compared to $RN50_{CLIP}$, we further explore using $ViT-B_{CLIP}$ as the GPS-BB when training a ResNet50 using VICReg and Barlow Twins. We present our results in Table 14. This shows that GPS-SSL depends on the GPS-BB and as the stronger the GPS-BB is, the stronger the performance of GPS-SSL will be on downstream tasks.

Table 14: Comparing test accuracy of GPS-SSL versus SSL after 400 epochs of training, when a $ViT-B_{CLIP}$ is used as the GPS-BB. We mark the best-performing score in **bold**.

| Method | GPS-BB | FGVCAircraft | PathMNIST | TissueMNIST |
|---|---|---|---|---|
| VICReg | — | 38.74 | 93.22 | **60.18** |
| GPS-VICReg | $ViT-B_{CLIP}$ | **70.03** | **93.48** | 58.60 |
| Barlow Twins | — | 18.12 | 92.03 | **61.69** |
| GPS-Barlow | $ViT-B_{CLIP}$ | **32.37** | **92.35** | 52.54 |

