# OpenReview forum: "GPS: Guided Positive Sampling in Self-Supervised Learning"
_TMLR — Rejected by TMLR_

### Review · Reviewer_SfRh · 2024-09-05

**Summary Of Contributions:**

The paper proposes a method for generating positive samples in contrastive self-supervised learning. The method generalizes the recently proposed nearest neighbor contrastive learning of representations (NNCLR) to use the neighbors in any semantic embedding space that the modeler chooses.

In experiments, the method is compared to other standard data augmentation methods for contrastive learning. A variety of SSL tasks are used in experiments.

**Audience:**

Yes

**Claims And Evidence:**

No

**Requested Changes:**

- Section 3.1: Typos in "diversity measure such that the rank of the embedings or proxies of their entropy."
- Eq 4: GPS selects the furthest point in the neighborhood, but is there a good justification for this? Why not a random point in the neighborhood?
- Proposition 1: I don't understand the claim that GPS-SSL simplifies to NNCLR when g_gamma=f_theta and tao approaches zero. Why is this true only when tao approaches zero, and not for all tao? Please clarify.
- Equation 5: I don't believe this claim, or at least I think it needs to be modified. It seems applying the DA twice, e.g. DA(DA(x; rho); rho), will not necessarily be in the dataset. This has implications for Theorem 1. I think the authors need to explain this more carefully (if it can be done).

**Strengths And Weaknesses:**

Strengths:
- The method is simple and practical. It allows the modeler to incorporate domain knowledge and outside information in the form of other pre-trained embeddings. It also can be combined with existing methods.
- The presentation is clear, including the connections to other SSL methods. The fact that it generalizes other known methods is very nice.
- Experiments were performed on multiple datasets, including datasets beyond the standard object classification task.
- Code is provided for the experiments.

Weaknesses:
- The method assumes that neighbors in some embedding are good positive examples for contrastive learning. The only evidence for this is empirical. No theoretical justification is provided.
- The method introduces the neighborhood size hyperparameter, tau, that must be tuned.

---

> ### Author Response · Authors · 2024-10-19
> **Part 1/1**
>
> - **The method assumes that neighbors in some embedding are good positive examples for contrastive learning. The only evidence for this is empirical. No theoretical justification is provided.**
>
> This is indeed an important observation. The current work relies on empirical evidence to demonstrate the effectiveness of using neighbors in the embedding space as positive examples. We acknowledge the need for theoretical justification and would explore this aspect in future work (Appendix 7.4) to provide a more comprehensive foundation for the approach.
>
> We note that previous studies have also demonstrated empirical benefits of using backbone embeddings as semantic-preserving spaces with Euclidean metrics. For example [this study](https://web.archive.org/web/20170610222724/http://www.ntu.edu.sg/home/egbhuang/pdf/ELM-Rosenblatt-Neumann.pdf) or [this study]( https://arxiv.org/abs/2407.08275) depict that property and validate it empirically. Unfortunately, we are not aware of theoretical studies that would prove that after training occurred. This is because training can distort what is being learned based on the optimizer, data, loss and theoretically characterizing that final solution remains a challenging open problem. Theory however can bring some answers with randomly initialized models with particular architectures, as with the [Neural Tangent Kernel](https://arxiv.org/abs/1806.07572) for example. However that setting involves an infinitely wide architecture that is therefore not trained, hence a very far setting from ours. We will however add that discussion in the paper if the reviewer deems it interesting.
>
> - **The method introduces the neighborhood size hyperparameter, tau, that must be tuned.**
>
> We appreciate this valuable point. We have enhanced our discussion in the revised version to clarify that $\tau$ is a function of $x$ based on the k-nearest neighbor distances. However, the selection of $k$ is indeed a hyperparameter, which we further describe in Appendix 7.3 concerning the search for optimal hyperparameters regarding $k$.
>
> - **Section 3.1: Typos in "diversity measure such that the rank of the embedings or proxies of their entropy."**
>
> Thank you for pointing this out. We have updated the corresponding text and corrected the typographical errors in the revised version.
>
> - **Eq 4: GPS selects the furthest point in the neighborhood, but is there a good justification for this? Why not a random point in the neighborhood?**
>
> We appreciate this feedback. In our actual implementation of GPS-SSL, we select a random point within the neighborhood. The mention of selecting the furthest point was a typographical error, which has been corrected in the revised paper.
>
> - **Proposition 1: I don't understand the claim that GPS-SSL simplifies to NNCLR when g_gamma=f_theta and tao approaches zero. Why is this true only when tao approaches zero, and not for all tao? Please clarify.**
>
> This clarification stems from NNCLR’s approach, where it selects the top nearest neighbor from its queue as the positive sample. When $\tau$ approaches zero in the GPS-SSL framework, it specifically ensures the selection of the top nearest neighbor, identical to NNCLR's strategy.
>
> - **Equation 5: I don't believe this claim, or at least I think it needs to be modified. It seems applying the DA twice, e.g. DA(DA(x; rho); rho), will not necessarily be in the dataset. This has implications for Theorem 1. I think the authors need to explain this more carefully (if it can be done).**
>
> Equation 5 illustrates an idealized scenario of data augmentations, proposed to show how GPS-SSL could theoretically substitute in an ideal case. The equation specifies the existence of $\rho_2$ such that $DA(DA(x; \rho_1); \rho_2)$ remains within the original dataset, noting that $\rho_1$ and $\rho_2$ are not necessarily equal. This distinction is crucial to understanding its application within Theorem 1.
>
> In addition, that assumption is quite realistic in many settings. That type of data model is known as assuming a "group structure on the data", i.e., where a repeated action of the group operator (here the DA) produces a new sample living in the same space. Numerous theory of deep learnings rely on that assumption, e.g., [this](https://arxiv.org/abs/2310.11366) or [this](https://openaccess.thecvf.com/content/CVPR2022/papers/MacDonald_Enabling_Equivariance_for_Arbitrary_Lie_Groups_CVPR_2022_paper.pdf). In practice, thinking of DA such as rotation and cyclical translations, we see that this assumption holds. However, it is true that this may not always hold based on the DA and dataset. We will make sure to mention that more clearly and also to emphasize that Theorem 1 and that analysis serves as motivation for GPS-SSL, not as guarantees.

---

> > ### Comment · Reviewer_SfRh · 2024-10-23
> >
> > I appreciate the authors clarifying the issues above.
> >
> > I agree that empirical results can be sufficient to motivate the use of a method like this, although I suspect there will be situations where it fails (e.g. when the embedding neighborhood is not good enough). I don't think this manuscript does enough to study the way in which the model can fail, which would be useful for practitioners wanting to use this method.

---

> > > ### Author Response · Authors · 2024-10-23
> > > **Response to SfRh**
> > >
> > > We thank the reviewer for their valuable feedback and for highlighting this important consideration.
> > >
> > > Regarding the quality of the GPS-BB, Table 1(a) illustrates various configurations, showing that performance improves as the GPS-BB performs better, underscoring that if a high-quality embedding space is not chosen, GPS-SSL may not perform optimally. Moreover, even with the strongest GPS-BBs in our experiments (e.g., CLIP_RN50 in Table 2 or CLIP_ViT_B16 in Table 14), GPS-SSL is either compatible or underperforms compared to the original SSL methods on medical datasets such as PathMNIST and TissueMNIST, further emphasizing this limitation.
> > >
> > > We have revised the manuscript to address this failure scenario in the “Limitations” subsection of Section 6.

---

### Review · Reviewer_kFPq · 2024-10-03

**Summary Of Contributions:**

This paper introduces Guided Positive Sampling (GPS), a flexible method designed to enhance existing SSL techniques by leveraging prior knowledge for improved positive sample selection. GPS defines positive samples through nearest neighbour sampling within a pre-trained representation. By utilizing this prior knowledge, GPS reduces the dependency on fine-tuning optimal data augmentation strategies for generating positive pair in SSL.

**Audience:**

Yes

**Claims And Evidence:**

No

**Requested Changes:**

Please kindly refer to the weaknesses stated above.

**Strengths And Weaknesses:**

### Strengths
- The proposed method is general and compactable with existing off-the-shelf SSL methods.
- Experimental results show that GPS does not rely on strong data augmentation (DA), and significantly outperforms the baselines in the RHFlipAug  settings.
- Writing is mostly clear and easy to follow.



### Weaknesses
- [critical] Despite utilizing strong pre-trained representations as a prior for guided positive pair sampling, the proposed method shows limited improvements over baseline SSL methods, as seen in Table 2, StrongAug settings. In some cases, e.g., PathMNIST, TissueMNIST, and Barlow Twins, it even underperforms. Furthermore, a primary motivation for SSL is to reduce reliance on optimal DAs for generating positive pairs. However, the effectiveness of GPS-SSL may still largely depend on the specific pre-trained backbone used, which serves as the prior for positive pair sampling. For instance, Tables 8, 12, and 13 present results using ViT-MAE as an alternative backbone to the CLIP ResNet-50 used in the main experiments, demonstrating that GPS-SSL's performance can be significantly influenced by different pre-trained backbones, sometimes underperforming compared to the corresponding baselines. Therefore, I think more comprehensive experiments are needed to assess the sensitivity of GPS-SSL to various GPS-BBs, analogous to how traditional SSL methods are tested with different data augmentation strategies. It would be beneficial to conduct these experiments with more SSL baselines beyond just SimCLR to further show that GPS-SSL is truly a flexible and versatile technique.

- [minor] Although GPS outperforms in the RHFlipAug settings, is using RHFlipAug ( i.e, *only* random horizontal flip for augmentation ) really a practical scenario for SSL or a common practice for comparison between different method - as most SSL methods (despite might be sensitive to the choice of augmentation strategies) would use multiple augmentations. That said, could the authors provide more discussion on why this an important case to consider, to better highlight the strength and motivation of the proposed GPS?

- [minor] In table 2, the performance of the proposed methods, in the strong augmentation settings, is quite inconsistently.  The performance seems to largely depend on the target dataset used for evaluation, and also the specific baseline SSL used with GPS, e..g, GPS-Barlow Twins mostly underperform comparing to Barlow Twins. It would be better if the authors could share some insights on these failure cases, (alternatively, why GPS seem to always work better with SimCLR as a SSL base algorithm, or on the aircraft dataset). I think this will strengthen the paper.

- [minor] It seems that there is a minor conflict in the description in eq (3) with the exactly implementation on how GPS defines the nearest neighbours region. E.g., \tau is used in eqn (3) which defines nearest neighbours based on a pre-defined threshold, but both the experimental results in Fig.5 and the writing that follows seem to suggest that k-nearest neighbours is employed - where k is a hyperparaemter -  regardless of the value of \tau? Although the two ideas are very similar, it would be better to be consistent.

---

> ### Author Response · Authors · 2024-10-19
> **Part 1/2**
>
> - **Despite utilizing strong pre-trained representations as a prior for guided positive pair sampling, the proposed method shows limited improvements over baseline SSL methods, as seen in Table 2, StrongAug settings. In some cases, e.g., PathMNIST, TissueMNIST, and Barlow Twins, it even underperforms.**
>
> The performance variability can be attributed to the nature of the datasets. Fine-grained datasets of more familiar objects, like Food101, Revised-HotelsID, and FGVCAircraft, show similar behavior. However, datasets like PathMNIST and TissueMNIST belong to different domains, where the CLIP-RN50 might not be the most suitable GPS-BB, leading to subpar performance in the StrongAug setting.
>
> In light of our new experiments, we observe that by using CLIP-ViT-B16 as the GPS-BB, as illustrated in Table 14, GPS-Barlow outperforms Barlow Twins on FGVCAircraft (32.37 vs. 18.12) and PathMNIST (92.35 vs. 92.03) in the StrongAug setting. Additionally, GPS-Barlow outperforms Barlow in Table 2 on Cifar10, even with a weaker GPS-BB ($RN50_{CLIP}$), achieving scores of 88.52 vs. 88.34.
>
> - **Furthermore, a primary motivation for SSL is to reduce reliance on optimal DAs for generating positive pairs. However, the effectiveness of GPS-SSL may still largely depend on the specific pre-trained backbone used, which serves as the prior for positive pair sampling. For instance, Tables 8, 12, and 13 present results using ViT-MAE as an alternative backbone to the CLIP ResNet-50 used in the main experiments, demonstrating that GPS-SSL's performance can be significantly influenced by different pre-trained backbones, sometimes underperforming compared to the corresponding baselines. Therefore, I think more comprehensive experiments are needed to assess the sensitivity of GPS-SSL to various GPS-BBs, analogous to how traditional SSL methods are tested with different data augmentation strategies. It would be beneficial to conduct these experiments with more SSL baselines beyond just SimCLR to further show that GPS-SSL is truly a flexible and versatile technique.**
>
> We thank the reviewer for this thoughtful point. Various GPS-BBs have already been tested with SimCLR in Table 1(a) as mentioned. Furthermore, CLIP ViT-B16 has been introduced as a new setting for mining positive samples with VICReg and Barlow Twins (Table 14). We observe that a stronger GPS-BB enhances the performance of GPS-SSL, as expected. Specifically, in Table 14, GPS-VICReg surpasses VICReg on FGVCAircraft (70.03 vs. 38.74) and PathMNIST (93.48 vs. 93.22), and GPS-Barlow exceeds Barlow Twins on FGVCAircraft (32.37 vs. 18.12) and PathMNIST (92.35 vs. 92.03).
>
> - **Although GPS outperforms in the RHFlipAug settings, is using RHFlipAug ( i.e, only random horizontal flip for augmentation ) really a practical scenario for SSL or a common practice for comparison between different method - as most SSL methods (despite might be sensitive to the choice of augmentation strategies) would use multiple augmentations. That said, could the authors provide more discussion on why this an important case to consider, to better highlight the strength and motivation of the proposed GPS?**
>
> The answer was provided in the general answer. This setting is beneficial for two reasons: firstly, it demonstrates the model's robustness to simple augmentations as opposed to meticulously crafted ones. Secondly, selecting optimal augmentations depends on user knowledge regarding the target dataset and the transformations that preserve image semantics. However, when dealing with a novel real-world dataset where such information might be scarce, relying on prior knowledge from a pre-trained network coupled with simple augmentations, like RHFlipAug, could be a more practical approach.

---

> > ### Author Response · Authors · 2024-10-19
> > **Part 2/2**
> >
> > - **In table 2, the performance of the proposed methods, in the strong augmentation settings, is quite inconsistently. The performance seems to largely depend on the target dataset used for evaluation, and also the specific baseline SSL used with GPS, e..g, GPS-Barlow Twins mostly underperform comparing to Barlow Twins. It would be better if the authors could share some insights on these failure cases, (alternatively, why GPS seem to always work better with SimCLR as a SSL base algorithm, or on the aircraft dataset). I think this will strengthen the paper.**
> >
> > We appreciate this observation. After further hyper-parameter tuning, we corrected the Barlow Twins value on FGVCAircraft in Table 2 (updated from 15.47 to 20.23). As mentioned in the “Strong Augmentation Experiments” section, the inconsistency highlighted by the reviewer stems from the relevance of GPS-DS to the target dataset. For example, since LAION-400M does not emphasize medical images and because CLIP-trained models were developed using LAION-400M, this GPS-DS is inadequate for PathMNIST and TissueMNIST, as illustrated in Table 2 and noted by the reviewer.
> >
> > - **It seems that there is a minor conflict in the description in eq (3) with the exactly implementation on how GPS defines the nearest neighbours region. E.g., \tau is used in eqn (3) which defines nearest neighbours based on a pre-defined threshold, but both the experimental results in Fig.5 and the writing that follows seem to suggest that k-nearest neighbours is employed - where k is a hyperparaemter - regardless of the value of \tau? Although the two ideas are very similar, it would be better to be consistent.**
> >
> > To simplify notations, the theory utilized $\tau$; however, we concur that this introduces an inconsistency. Although the two approaches are similar, as noted by the reviewers, they can be made precisely equivalent by considering $\tau$ as a function of $x$ (based on the k-NN distance for each sample). We will implement this change in the theory section for consistency and mention in the experiments that $\tau(x)$ is selected from the k-NN distances.

---

> ### Comment · Reviewer_kFPq · 2024-10-24
>
> Thank you for the detailed rebuttal and revisions. I have reviewed the updated manuscript and appreciate the effort in addressing my concerns. My concerns are partially addressed. Please see details below:
>
>
>
> ### Regarding Additional GPS-BB Experiments
> > **The performance variability can be attributed to the nature of the datasets.** ... We thank the reviewer for this thoughtful point. Various GPS-BBs have already been tested with SimCLR in Table 1(a) as mentioned. Furthermore, CLIP ViT-B16 ....
>
> > In light of our new experiments, we observe that by using CLIP-ViT-B16 as the GPS-BB, as illustrated in Table 14, ... **the inconsistency highlighted by the reviewer stems from the relevance of GPS-DS to the target dataset.**
>
> I appreciate the additional experiments with different GPS-BBs. I think the results support my argument that the effectiveness of GPS heavily depends on selecting a suitable GPS-BB for a particular dataset. This is akin to having prior user knowledge about the dataset, for choosing appropriate augmentation strategies before training. For instance, in Table 14, even with strong backbones, performance on TissueMNIST can be worse than the baselines, with a decline of up to -9.1% for GPS-Barlow Twins + ViT-CLIP, which is even worse than the -4.6% seen in Table 2 with GPS-Barlow Twins + RN50-CLIP, a weaker backbone. While there is significant improvement on the FVGA dataset with GPS, the inconsistent performance across datasets suggests that GPS does not consistently yield better results, and stronger backbones do not always lead to better outcomes.
>
> Nevertheless, I think that inconsistency and dependence on target dataset/backbone is less of an issue if the authors moderate their claim that "GPS overcomes heavily tuning of DAs," considering that: 1) GPS still requires DAs, and 2) performance remains inconsistent and sensitive to prior knowledge. I view the main contribution of GPS as an empirical study proposing alternative positive pair sampling strategies, rather than a method that mitigates sensitivity to prior selection or strives for SOTA performance, as noted in the conclusion.
>
> That said, I also strongly encourage the authors to shed some lights on what exactly makes a good prior knowledge for positive fair sampling, or an detailed analysis of the failure cases of GPS, as I have mentioned in the weaknesses above (I don't request the authors to do so). I acknowledge that the authors have attempted to explain these in the rebuttal though dependence on dataset but I find this analysis is still too coarse for bringing valuable insights. While I understand that such analysis may be beyond this paper's scope, it could significantly enhance the paper's quality and contribution.
>
> ### Regarding Weak or Strong DAs
> >Secondly, selecting optimal augmentations depends on user knowledge regarding the target dataset and the transformations that preserve image semantics. However, when working with a novel real-world dataset where such information might be limited,...
>
> I understand that selecting optimal DAs is challenging and relies on prior user knowledge about the target dataset. However, my point is why strong DAs cannot be the default strategy, given that: 1) they generally yield better performance compared to RHFlipAug, and 2) they are a standard practice. If so, wouldn't this render RHFlipAug less practical since strong DAs would be applied by default?
>
> >...relying on prior knowledge from a pre-trained network combined with simple augmentations, like RHFlipAug, could be a more effective and practical choice.
>
> Why is "a pre-trained network combined with simple augmentations, like RHFlipAug" a "more effective and practical choice"? Especially since Table 2 shows that strong DAs still perform better on average compared to this proposed strategy?
>
> I hope these suggestions help clarify and enhance the manuscript's contribution.

---

### Review · Reviewer_znsU · 2024-10-09

**Summary Of Contributions:**

This paper introduces a new method for Self-Supervised Learning (SSL) called Guided Positive Sampling (GPS-SSL). This method aims to incorporate prior knowledge into SSL by improving the selection of positive samples, moving away from the reliance on data augmentations (DA) traditionally used in SSL methods. GPS-SSL is especially suitable for the situation when the datasets are under-explored and the DA of the certain dataset is not well studied. In summary, GPS-SSL aims to improve this process by leveraging prior knowledge to select more meaningful positive pairs. Instead of relying solely on augmentations, GPS-SSL finds positive samples through a nearest-neighbor search in a predefined metric space, which can be informed by prior knowledge from pre-trained models or handcrafted embeddings.

**Audience:**

Yes

**Claims And Evidence:**

Yes

**Requested Changes:**

1. Where are the proofs of Proposition 1 and Theorem 1? Besides, you should use Proposition 1 and Theorem 1 instead of proposition 1 and theorem 1 in the main text of the paper.

2. What is $f_{\theta}$ in equation (1)?

3. In (4), the $x'$ is not unique, right? You will have many data points to achieve this maximal value.

4. In (5), can you explain more about the notation of configuration $\rho$?

5. In Figure 5 of the ablation over different values of $k$ with GPS-SimCLR on FGVCAircraft, can you explain why it is not a monotonic function of $k$?

6. Have you tried some specific medical datasets in your simulations?

**Strengths And Weaknesses:**

## Strengths:

GPS-SSL introduces a new way of selecting positive samples using guided sampling rather than relying solely on data augmentations. This approach addresses the limitations of standard SSL methods and opens up a new direction for improving representation learning.
The method alleviates the need for heavily tuned data augmentation recipes, which can be challenging to design for specialized datasets. GPS-SSL shows significant performance improvements on under-explored datasets where effective data augmentation is not well established. Additionally, this method can be easily integrated with various existing SSL frameworks (e.g., SimCLR, BYOL, VICReg), making it versatile and practical.

## Weaknesses

While the empirical validation covers a range of datasets, most experiments focus on image classification tasks. It would be interesting to see how GPS-SSL performs on other modalities or datasets from other domains, such as text, audio, or multi-modal data, to generalize the applicability of the proposed approach.

Some parts of the theoretical sections, especially when introducing the equations, can be challenging to follow. Providing more intuitive explanations and explaining the notations would improve the readability of the theoretical justifications.

The paper could benefit from additional ablation studies investigating how different qualities of embeddings (e.g., embeddings from a weakly trained model versus a strong pre-trained model) impact GPS-SSL performance. This would help quantify the relationship between embedding quality and the benefits provided by the GPS-SSL approach.

---

> ### Author Response · Authors · 2024-10-19
> **Part 1/2**
>
> - **While the empirical validation covers a range of datasets, most experiments focus on image classification tasks. It would be interesting to see how GPS-SSL performs on other modalities or datasets from other domains, such as text, audio, or multi-modal data, to generalize the applicability of the proposed approach.**
>
> This is an excellent suggestion. Owing to the adaptability of GPS-SSL, we plan to extend our research into multi-modal domains, such as image-language, as part of future efforts. We also aim to explore domains where augmentations may not be well-defined, like language. Additionally, Section 7.4 has been included to elaborate on these future exploration plans.
>
> - **Some parts of the theoretical sections, especially when introducing the equations, can be challenging to follow. Providing more intuitive explanations and explaining the notations would improve the readability of the theoretical justifications.**
>
> We have added additional notations and clarifications to the equations. For example, we included explanations for notations like $f_{\theta}$ and $L_{SSL}$ in Section 3.1, and the notation $\tau$ has been explained in further detail. Please inform us if any confusions remain.
>
> - **The paper could benefit from additional ablation studies investigating how different qualities of embeddings (e.g., embeddings from a weakly trained model versus a strong pre-trained model) impact GPS-SSL performance. This would help quantify the relationship between embedding quality and the benefits provided by the GPS-SSL approach.**
>
> With the time constraints of the rebuttal period, we were able to add an additional set of embeddings using CLIP ViT-B16, trained with VICReg and Barlow Twins for two datasets under the StrongAug setting in Table 14.  Furthermore, Table 1 (a), depicting the different GPS-BBs that were tested and compared for GPS-SSL, has been updated accordingly. We have also added tot he Conclusion section the relationship between the prior knowledge and GPS-SSL as the reviewer kindly pointed out.
>
> - **Where are the proofs of Proposition 1 and Theorem 1? Besides, you should use Proposition 1 and Theorem 1 instead of proposition 1 and theorem 1 in the main text of the paper.**
>
> We appreciate your observation regarding the formal proofs for Proposition 1 and Theorem 1. At present, the claims made in these propositions rely on empirical observations. While formal proofs are not yet available, we have found these propositions to hold true when comparing GPS-SSL to other SSL methods.
>
> As for Theorem 1, standard SSL involves picking a sample ($x_0$) and applying two DAs to it to produce the positive pair. Our goal in this theorem is to show that we recover that exact setting when g is invariant to the DA being employed and for a small $\tau$. To see that, notice that if g is invariant to the DA employed, then $g(DA(x_0,\rho))=g(x_0)$ for any \rho. Hence, since those are equal, the search space over the nearest neighbor ($B(x_0)$) will only include the different views of $x_0$ -- therefore recovering the SSL setting.
>  We recognize the importance of theoretical validation and aim to develop formal proofs in future work in Appendix 7.4. Also, we have also ensured the capitalization of Proposition 1 and Theorem 1 in the revised manuscript.
>
> - **What is f_theta in equation (1)?**
>
> We apologize for the oversight. $f_{\theta}$ represents the SSL network being trained, and this has been clarified in the updated submission.
>
> - **In (4), the $x’$ is not unique, right? You will have many data points to achieve this maximal value.**
>
> Equation (4) has been revised in the updated submission. The point $x'$ is randomly selected from the set of nearest neighbors of point $x$.
>
> - **In (5), can you explain more about the notation of configuration $\rho$?**
>
> We thank the reviewer for raising that question and apologize for missing to clarify $\rho$ in our original submission and we have clarified it in our revised submission. $\rho$ is the parameter that determines the DA transformation applied onto the input. For example, consider rotation as the DA, rho would be a single scalar defining the amount of degrees to rotate the image for. If the DA is the vertical and horizontal translation, then who would be the horizontal and vertical shifts to apply.

---

> > ### Author Response · Authors · 2024-10-19
> > **Part 2/2**
> >
> > - **In Figure 5 of the ablation over different values of k with GPS-SimCLR on FGVCAircraft, can you explain why it is not a monotonic function of $k$?**
> >
> > For any sample $x$ in the dataset, the similarity of the $k$th nearest-neighbor to $x$ decreases as $k$ grows, adding more variance to the "positive pairs." However, adding excessive variance to the set of positive-pairs may introduce too much noise into the learning process, leading to the performance not being a monotonic function of $k$. If the GPS-DS (the dataset used for training the GPS-BB) has a comparable data distribution to the target dataset, the suitable number of $k$ is greater than when the target dataset diverges from the GPS-DS distribution.
> >
> > - **Have you tried some specific medical datasets in your simulations?**
> >
> > We conducted evaluations on TissueMNIST and PathMNIST but did not employ a GPS-BB pre-trained on medical datasets. We hypothesize that utilizing a GPS-BB trained on medical data would enhance GPS-SSL's performance on TissueMNIST and PathMNIST compared to using CLIP_RN50 as the GPS-BB.

---

> > ### Comment · Reviewer_znsU · 2024-11-15
> > **Official Comment by Reviewer znsU**
> >
> > Thanks for your rebuttal and detailed explanations. Most of my concerns have been addressed. However, for Proposition 1 and Theorem 1, if the authors cannot provide proof for these results, it would be better to write these as conjectures or observations.

---

### Author Response · Authors · 2024-10-19
**General Answer**

First and foremost, we would like to express our appreciation to all the reviewers for their thorough reviews. We are delighted to see that **all reviewers agree with us on the importance to develop Self-Supervised Learning (SSL) solutions that can learn from weak and non-optimal data-augmentations, and on the ability of GPS-SSL to take a step in that direction using prior knowledge**, e.g., “*The method alleviates the need for heavily tuned data augmentation recipes, which can be challenging to design for specialized datasets.*” (znsU), “*GPS reduces the dependency on fine-tuning optimal data augmentation strategies for generating positive pair in SSL*” (kFPq), “*It allows the modeler to incorporate domain knowledge and outside information in the form of other pre-trained embeddings*” (SfRh).

Moreover, we are pleased that **all reviewers found our empirical validation to be comprehensive and persuasive concerning weak data augmentations**; e.g., “*GPS does not rely on strong data augmentation (DA),*” (kFPq), “*GPS-SSL shows significant performance improvements on under-explored datasets where effective data augmentation is not well established,*” (znsU), and “*Experiments were performed on multiple datasets, including datasets beyond the standard object classification task.*” (SfRh).

Finally, we are gratified that **all reviewers agree on the practical usefulness of GPS-SSL, noting its compatibility with existing SSL methods.** As noted, “*Additionally, this method can be easily integrated with various existing SSL frameworks (e.g., SimCLR, BYOL, VICReg), making it versatile and practical.*” (znsU), “*The proposed method is general and compatible with existing off-the-shelf SSL methods.*” (kFPq), and “*The method is simple and practical. [...] It also can be combined with existing methods.*” (SfRh).

We briefly summarize our contributions below:

We introduce an extension of NNCLR, termed GPS-SSL, a Self-Supervised Learning solution offering two main advantages:

- An improvement over NNCLR that includes:

    - The removal of the queue and a simplified/generalized learning strategy.
    - Enhanced robustness against weak augmentations compared to NNCLR.

- An advancement for SSL in general:
    - A simplified approach to incorporate a priori knowledge into positive pair sampling for SSL training, enabling the use of any pretrained model to sample positive pairs.
    - The universality of the GPS recipe, applicable out-of-the-box across various SSL methods, such as VICReg, BarlowTwins, SimCLR, MoCo v3, and BYOL.

Our method and claims undergo thorough validation across numerous datasets and architectures, particularly in light of the additional experiments conducted during this rebuttal. These experiments demonstrate that the method significantly surpasses NNCLR and other SSL baselines when employing weak data augmentations. With strong (optimal) data augmentation, all methods yield similar performances on most datasets; however, on Food101 and FGVCAircraft, GPS-SSL performs better. This serves as a sanity check for full transparency; nonetheless, if optimal data augmentations are known, the benefit and motivation for using NNCLR or GPS-SSL over traditional positive pair generation would be significantly reduced.

The reviewers also raised a few key concerns:

1. **Relevance of RHFlipAug:** We thank the reviewers for highlighting this important issue. This setting is beneficial for two reasons: firstly, it demonstrates the model's robustness to simple augmentations as opposed to carefully designed ones. Secondly, selecting optimal augmentations depends on user knowledge regarding the target dataset and the transformations that preserve image semantics. However, when working with a novel real-world dataset where such information might be limited, relying on prior knowledge from a pre-trained network combined with simple augmentations, like RHFlipAug, could be a more effective and practical choice.

2. **Need for additional GPS-BB studies:** We appreciate the reviewers' suggestion to expand the ablation studies. During the limited rebuttal period, we trained and evaluated an RN50 using a CLIP-trained ViT-B16 as the GPS-BB for VICReg and Barlow Twins on FGVCAircraft, PathMNIST, and TissueMNIST, as shown in Table 14. The results indicate that a CLIP-trained ViT-B16 outperforms a CLIP-trained RN50 (Table 2) due to its superior performance. In this scenario, GPS-Barlow surpasses or matches Barlow Twins, and GPS-VICReg outperforms VICReg on FGVCAircraft and PathMNIST. This ablation study further demonstrates that improved GPS-BBs lead to better positive samples, resulting in enhanced GPS-SSL performance.

We firmly believe that these revisions have strengthened our submission. Please find our updated revised paper. We invite the reviewers to engage with us during the discussion period if any issues remain unresolved. We would greatly appreciate the reviewers acknowledging our rebuttal in their assessments.

---

### Decision · Action_Editor_rdZU · 2024-12-02

**Recommendation:** Reject

**Comment:**

Please see "claims and evidence" section.

**Audience:**

Appropriate.

**Claims And Evidence:**

The paper presents Guided Positive Sampling (GPS) for self-supervised learning (SSL), which can be integrated with existing methods. The core contributions are detailed in Section 3. While the reviewers appreciate the concept, they unanimously agree that the paper requires major revisions to be considered a significant contribution. There are several issues identified:

1. Theoretical contributions are missing. Although propositions and theorems are included, the proofs are absent. This has been highlighted by the reviewers, yet no corrections have been made, with the authors stating it as future work. If that is the case, then it is better to remove them entirely and replace them with empirical observations or remarks. A bigger concern is that without any proper theoretical justification the entire paper looks like an empirical exercise and not really a solid work.

2. While Section 3 is the main contribution of the paper, its motivation is largely missing, with only an afterthought explanation provided. Ideally, there should be a strong motivation as to why consider those steps.

Therefore, the claims and evidence do not align, and further revisions are necessary to bring the paper to a publishable state.

**Resubmission Of Major Revision:**

The authors may consider submitting a major revision at a later time.